# Roles of Auxin in the Growth, Development, and Stress Tolerance of Horticultural Plants

**DOI:** 10.3390/cells11172761

**Published:** 2022-09-05

**Authors:** Qiongdan Zhang, Min Gong, Xin Xu, Honghai Li, Wei Deng

**Affiliations:** Key Laboratory of Plant Hormones and Development Regulation of Chongqing, School of Life Sciences, Chongqing University, Chongqing 400044, China

**Keywords:** auxin, growth, development, stress, horticultural plants

## Abstract

Auxin, a plant hormone, regulates virtually every aspect of plant growth and development. Many current studies on auxin focus on the model plant Arabidopsis thaliana, or on field crops, such as rice and wheat. There are relatively few studies on what role auxin plays in various physiological processes of a range of horticultural plants. In this paper, recent studies on the role of auxin in horticultural plant growth, development, and stress response are reviewed to provide novel insights for horticultural researchers and cultivators to improve the quality and application of horticultural crops.

## 1. Introduction

Horticultural crops, including fruit trees, vegetables, flowers, herbs, and ornamental plants, are essential to daily life. With the development of the times and the progress of society, horticultural plants not only play an economic role in providing food for humans and animals but are also playing an increasingly important social role in influencing human lifestyle, shaping human culture, and beautifying the human living environment [1]. These changes in the functions of horticultural plants mean that they occupy an increasingly significant place in human life, encouraging more people to invest in the research of horticultural plants, to produce more varieties and better products, and to improve the practical application of horticultural plants.

The phytohormone auxin coordinates many critical processes in plant growth, development, and adaptation to the environment [2]. Auxin functions are associated with specific biosynthesis, homeostasis, transport, and signal transduction pathways [3]. For the model plant Arabidopsis thaliana, and for some field crops, such as wheat, maize, rice, etc., research on the role of auxin in their growth, development, and stress response has been in-depth and extensive [4,5,6,7]. However, research on auxin in horticultural plants remains scarce. The reason is possibly due to the wide variety of horticultural plants and the many horticultural plants with unique physiological structures. It is precisely because of the abundance of horticultural plant resources and their many physiological processes that differ from model plants and field crops that the role of auxin in these plants is worthy of our further research and exploration, expanding and deepening people’s understanding and thinking about this plant hormone. This paper reviews the recent research on the role of auxin in the growth, development, and stress response of some horticultural plants. It is expected to expand our understanding of the mechanism of auxin regulation of plant growth and development, and to provide ideas to help horticultural researchers to obtain normal growth and development, high yield, and high-quality horticultural plants.

## 2. A Brief Overview of Auxin

The activity of auxin in various physiological processes can be spatiotemporally regulated through three main regulatory measures: auxin biosynthesis and inactivation, auxin-directed transport, and signal transduction.

IAA (Indole-3-acetic acid) is a ubiquitous endogenous auxin in plants, synthesized by both Trp (tryptophan)-independent and Trp-dependent pathways [8,9]. To date, however, only one complete auxin biosynthetic pathway has been identified in plants—the tryptophan (Trp)-dependent pathway: TAA/YUC (TRYPTOPHAN AMINOTRANSFER-ASE OF ARABIDOPSIS/YUCCA) [10,11]. The TAA/YUC pathway converts Trp to IAA through two sequential chemical steps. Initially, the TAA family of aminotransferases metabolizes Trp into IPyA (indole-3-pyruvate). Next, the YUC family of flavin-containing monooxygenases catalyze the oxidative decarboxylation of IPyA to produce IAA. It is the major auxin biosynthetic pathway and is indispensable for all significant developmental processes [12].

The auxin response is concentration-dependent in most tissues, with different tissues having distinct responses to varying amounts of exogenous auxin [13]. Higher auxin concentrations tend to be inhibitory, so the optimal endogenous levels must be firmly controlled. Multiple mechanisms exist for regulating auxin homeostasis; these include dynamic biosynthesis, degradation, transport, and conjugate formation of free IAA [14]. IAA is plants’ primary form of active auxin [2,15]. A conserved mechanism for maintaining IAA homeostasis in monocotyledonous and dicotyledonous plants is through the conversion of free IAA to its conjugated form. The family of GH3 (Gretchen Hagen 3) proteins has auxin amino acid synthase activity. It converts active IAA into an inactive state by binding free IAA to amino acids. The combination and degradation of amino acid molecules and IAA help maintain IAA’s homeostasis in plants [16]. GH3 plays a role in the negative feedback regulation of IAA concentration, and excess IAA upregulates *GH3* expression, resulting in the storage or degradation of amino-acid-bound IAA. The *GH3* gene is essential for average plant growth and adaptation to environmental stress [14,17].

Auxin regulates many physiological processes by controlling gene transcription through the SCF^TIR1/AFB^-Aux/IAA-ARF nuclear signaling module. The module requires an ensemble of three key constituents. The F-box proteins AFB1-AFB5/TIR1 are auxin receptor subunits of SCF-Family E3 ubiquitin, which ligases the activator class of ARF (AUXIN RESPONSE FACTOR) transcription factors and the transcriptional repressors Aux/IAA [18]. Auxin perception begins with auxin binding to TIR1/AFB receptors. It leads to the degradation of Aux/IAA proteins that physically interact with ARF transcription factors and inhibit auxin signaling [19]. Auxin stimulates the ubiquitination and degradation of Aux/IAAs protein and releases the transcriptional activity of ARF to turn on the transcription of downstream auxin-responsive genes. Different developmental processes are regulated by different Aux/IAA-ARF modules and corresponding auxin response genes [20].

In higher plants, auxin has two distinct but interconnected transport systems: the first is a fast, non-directional flow with photoassimilates in the phloem, and the second is a slow and directed intercellular polar auxin transport (PAT) [21]. Polar transport is the directional active transport process of auxin molecules in plant tissues, which depends on specific carrier proteins to complete. The combined activity of auxin influx and efflux carrier proteins produces local hormone maxima. The directional auxin gradients are indispensable for essential developmental processes of plants, such as organ development, apical hook formation, gravitropism and hydrotropism (bending to directional root growth), and phototropism [22]. Auxin polar transport depends on three transport proteins: the import carrier protein AUX/LAX (AUXIN1/LIKE-AUX1) family, the export carrier protein PIN (pin-formed) family, and the carrier protein ABCB/MDR/PGP (ATP binding cassette B/Multidrug-resistance/p-glycoprotein) family with both import and export functions [23] (Figure 1).

## 3. The Roles of Auxin in Vegetative Growth of Horticultural Plants

### 3.1. The Role of Auxin in Root Development of Horticultural Plants

Plant roots are an essential part of plant structure and are associated with plant fixation, water, and nutrient absorption. Plant root systems are generally composed of primary roots (PRs) and lateral roots (LRs) [24]. In contrast, lateral roots initiation to shoot branching occurs through the endogenous formation of new primordium that grows out of primary roots or other tissues. Adventitious roots (ARs) develop from non-root tissues of plants (including leaves, stems, hypocotyls, and reproductive organs), and are prevalent in both monocotyledonous and dicotyledonous plants [25].

Trees and flowers are vital for horticultural construction, and for current tree and flower breeding and commercialization, vegetative propagation is the most important means to preserve and reproduce excellent individuals. The formation of ARs is a key factor in the efficiency of clonal propagation and explant survival [26]. Auxin is one of the most indispensable phytohormones that modulate the formation of LR and AR [27,28]. The AR formation process is associated with relatively high IAA (indoleacetic acid) content. This process can be divided into two parts: a series of founder cell divisions and the elongation of the interfascicular cambium adjacent to the vascular tissue [29,30]. Some researchers have found that treating apple cuttings with exogenous auxin promotes the cell divisions mentioned above. N-1-Naphthylphthalic acid (NPA) is an inhibitor of the polar transport of auxin. The opposite result occurred when NPA was used to treat apple cuttings, which resulted in abnormal cell divisions during the early stage of AR formation [31]. Members of the GH3 (Gretchen Hagen 3) protein family play a role in auxin homeostasis, which is an essential regulator of plant development. A vegetative propagation study of axillary stem cuttings in new carnation (*Dianthus caryophyllus*) found that enhanced conjugation of auxin by GH3 enzymes leads to poor adventitious rooting [32]. Similarly, overexpression of apple (*Malus Domestica*) *MsGH3.5* markedly reduced free IAA content, and its *MsGH3.5* overexpressing lines produced fewer ARs than controls [33]. In addition, it has been reported that eight members of the apple Auxin Efflux Carrier PIN family are all expressed during the formation of AR. *MdPIN8* and *MdPIN10* are up-regulated during AR induction, and *MdPIN3* is up-regulated early in AR initiation, while *MdPIN7* and *MdPIN2* are up-regulated late in initiation, and *MdPIN4*, *MdPIN5*, and *MdPIN8* are up-regulated during AR elongation. This suggests that different MdPIN family members may be involved in various stages of AR formation in different ways [31]. Post-translational modification of ARFs and AUX/IAAs, including phosphorylation, ubiquitination, acetylation, and SUMOylation, also significantly regulates auxin signaling. SUMOylation is a significant posttranslational modification that controls numerous regulatory processes, including transcriptional activity, protein–protein interactions, degradation, and localization [34]. Recent studies show that SUMOylation of apple ARFs can regulate lateral root development [35]. BT proteins belong to BTB-TRANSCRIPTION ADAPTOR PUTATIVE ZINC FINGER (TAZ) domain proteins, which are plant-specific BTB proteins. BTB protein is involved in the ubiquitination and degradation process. The apple BTB protein MdBT2 inhibits apple AR formation by interacting with MdARF8 and MdIAA3 [36].

Many studies have shown that sugar signals play a crucial role in root growth and development. For example, the study found that under the treatment of exogenous sucrose, the total volume and surface area of peach (Prunus persica) roots increased, and the growth and number of LR were increased. This is because peach PpSnRK1 (sucrose non-fermentation-1-protein kinase-1) is activated during this process and can interact with PpIAA12/PpPIN-likes6 to promote auxin accumulation in roots in response to sucrose, thereby regulating roots growth [37]. The waterlogged parts of plants form new ARs, which contain more aerenchyma than primary roots [38]. It can promote plant gas exchange and water and nutrient uptake and improve plant adaptation to waterlogging [39]. For example, cucumber (*Cucumis* sativus L.) can survive waterlogging by producing adventitious roots (ARs) to promote gas exchange. The germination of AR in cucumber under water stress is regulated by the interaction of sugar and auxin [40] (Figure 2).

### 3.2. The Role of Auxin in Shoot Development of Horticultural Plants

Shoot branching is a significant horticultural trait that seriously affects plant structure and crop yield. In flowering plants, the lateral branches are formed by the axillary meristem (AM) of the leaf axils. Then, AMs develop into axillary buds composed of several leaf primordia and a meristem. At this stage, they either grow out to form a branch or choose to remain dormant [41]. Excessive shoots compete for light harvesting and nutrient allocation, often negatively impacting crop yields. Therefore, inhibition of above-ground branching in plants has become a key selection trait during crop domestication [42]. TB1 (TEOSINTE BRANCHED1)/BRC1 (BRANCHED1) is an integrator of phytohormonal, nutritional and environmental signals that act in axillary buds to inhibit lateral shoot growth. Auxin stimulates strigolactone (SL) biosynthesis during this process, resulting in increased expression of *TB1/BRC1* to inhibit branching in pea plants. On the contrary, auxin inhibits cytokinin (CK) biosynthesis to promote transcription of *TB1/BRC1* and repress shoot branching in pea plants [43]. In addition, cucumber BRC1 (CsBRC1) can directly bind to CsPIN3 and negatively regulate its expression, resulting in the accumulation of auxin in axillary buds and hiding the growth of axillary buds [44]. In tomatoes, inhibition of *SlPIN4* and *SlPIN3* expression was found to alter shoot structure [45]. This suggests that preventing auxin export from dormant buds by inhibiting PINs would block their outgrowth. In addition, overexpression of the *Gretchen hagen3* (*GH3*) gene *MsGH3.5* in wild apples (*Malus sieversii* Roem) also inhibits the growth of plant stems, resulting in dwarf phenotypes [33] (Figure 2).

### 3.3. The Role of Auxin in Leaf Development of Horticultural Plants

The shape characteristic of plant leaves is an essential horticultural trait. The growth characteristics of leaves determine the diversity of their shapes. The leaf is formed at the stem apical, and the growing point of the stem apical is called the shoot apical meristem (SAM) [46]. Auxin is a critical signaling factor in the regulation of organogenesis at the SAM [47,48]. It is considered to be a key determinant of environmentally induced changes in leaf shape within species because leaf-shape changes are regulated by many auxin-related genes, such as the auxin receptor gene *TIR1*, the auxin response factor genes (*ARFs*), the auxin importer genes (*AUXs/LAXs*), and the pin-formed auxin exporter genes (*PINs*) [48,49,50].

PIN1-directed auxin transport is a significant regulator of leaf development and leaflet initiation [51,52]. A study of tomato PIN1 found that auxin transport to the inner layer of leaves may be required for leaflet initiation [53]. An apple basic/helix–loop–helix transcription factor *MdbHLH3* is expressed at a higher level at a higher altitude. It controls leaf shape by specifically activating auxin amido conjugate synthase gene *MdGH3-2* and suppressing pin-formed auxin efflux transporter gene *MdPIN1* [54]. The NAC transcription factor VviNAC33 of grapevine (*Vitis vinifera* L.) is a negative regulator of the auxin pathway. VviNAC33 is involved in inhibiting cell division during leaf development by directly binding to *the PIN1* promoter and inhibiting its expression [55].

Trichomes are a typical structure in the epidermis of terrestrial plants, which are widely distributed in the stems, leaves, and flowers of plants. It is a natural protective barrier against natural hazards, such as herbivores, pathogen attacks, ultraviolet (UV) exposure, and excessive transpiration [56,57,58]. Recent studies have shown that auxin signaling factors SlARF3, SlARF4, and SlIAA15 are involved in the formation of type II, V, and VI trichomes in tomatoes, indicating that the initiation of trichomes requires auxin-dependent transcriptional regulation [59,60,61] (Figure 2).

## 4. The Roles of Auxin in Reproductive Growth of Horticultural Plants

### 4.1. The Role of Auxin in Flower Development of Horticultural Plants

The process of plant flowering involves many physiological processes, metabolic pathways, and gene regulatory mechanisms, representing the transition of plants from vegetative growth to reproductive growth [62,63]. Recently, it has been reported that strawberry FveARF4 can bind to the promoters of the floral meristem recognition genes *APETALA1 (AP1)* and *FULL (FUL)* to induce their expression, thereby promoting the flowering of woodland strawberries. It suggests that auxin is involved in the flowering pathway of strawberries by regulating the expression of *FveARF4* [64]. ARF is not only interested in the regulation of plant flowering but also in the regulation of flower abscission. RhARF7 is involved in regulating petal abscission in early rose development [65]. In addition, some transcription factors can modulate local auxin distribution by regulating the expression levels and patterns of *PIN* and *YUC* genes to control floral organ development. For example, transcriptional control of local auxin distribution by the CsDFB1 (cystatin-like protein DEFORMED FLORAL BUD1)-CsPHB (HD-ZIP III transcription factor PHABULOSA) module regulates floral organogenesis in cucumbers. It is because CsDFB1 impairs the CsPHB-mediated transcriptional regulation of *CsYUC2* and *CsPIN1* and thus plays an essential role in auxin distribution [66]. Furthermore, PIN1 and YUCCA1 were found in the developing flower head in Asteraceae flower head formation studies, suggesting that they play an active role in establishing auxin distribution in *M. inodora* capitulum [67] (Figure 3).

### 4.2. The Role of Auxin in Fruit Development of Horticultural Plants

In managing horticultural plants, fruit quality is the key to economic efficiency and horticultural effects. Auxin signaling is tightly regulated during fruit set and fruit development. Many plant hormones regulate the complex process of ovary-to-fruit transformation, of which the regulation of auxin predominates. For instance, ChARF3 can regulate ovule development and ovarian initiation in hazelnut (*C. heterophylla*). It acts by mediating the expression of a series of genes related to auxin biosynthesis and transport, cell division and proliferation, and flower and fruit development [68]. Cotton (*Gossypium hirsutum*) *GhARF2* and *GhARF18* genes may become vital regulators of cotton seed fiber cell initiation by regulating the expression of several transcription factor genes [69]. In tomatoes, SlARF9 negatively regulates cell division during early fruit development. Transgenic plants with elevated *SlARF9* mRNA levels formed smaller fruits than wild-type fruits due to reduced cell division activity. In contrast, transgenic plants with reduced *SlARF9* mRNA levels formed larger fruits than wild-type ones due to increased cell division activity [70].

Parthenocarpy has been recognized as an ideal agronomic trait to help growers to overcome the problem of low fruit yield under adverse environmental conditions, because when a parthenocarpic fruit is formed, its ovary does not need pollination and/or fertilization [71]. In addition, consumers also tend to choose parthenocarpic fruits, since the resulting fruit is seedless. RNAi silencing of the eggplant *SmARF8* gene causes parthenocarpy [72]. Similarly, transgene expression of an aberrant *AtARF8-4* also causes parthenocarpic tomato [73].

In contrast, mutations in *FveARF8* in strawberries did not result in a parthenocarpic fruit set. Instead, they resulted in larger fruit with enhanced sensitivity to auxin and GA treatments, suggesting that FveARF8 negatively regulates fruit growth effects [74], unlike SmARF8 and SlARF8, which played a positive role in regulating fruit growth. In fact, silencing of *SlPIN4* in tomatoes also leads to parthenocarpy, suggesting a role for pin-formed (PIN) protein in auxin regulation of fruit set rate [75].

Fruit weight or size is one of the main factors influencing commodity value and fruit quality. A candidate loci associated with loquat fruit weight through the integration of genomics, transcriptomics, and metabolic profiling was identified and revealed a vital role for auxin in regulating the fruit enlargement stage [76]. Silencing of *SlIAA17* in tomatoes results in increased fruit cell size and thicker pericarp, showing a phenotype with larger fruit than the wild type [77]. Phenotype analysis of fruit size over multiple seasons in two apple (*Malus ×domestica*) mapping populations observed a QTL that mapped to a region containing an auxin-responsive factor (*ARF106*). The gene was found to be expressed during the cell division and cell expansion stages, and may have a potential role in controlling fruit size [78]. Artificial microRNA (amiRNA) is an effective strategy to silence endogenous genes based on the structure of natural microRNA. After silencing the *SlARF5* gene using amiRNA technology, the fruit of the transgenic line showed parthenocarpic fruit. Compared with the wild type, the fruit of the transgenic line had fewer locular tissues, and the fruit size and weight were decreased [79]. Fruit size is under intensive selection during crop diversification or domestication, and fruit size is generally determined by various factors, including fruit length, diameter, or aspect ratio. [80]. Fruit length is an essential agricultural trait for melon vegetables, such as cucumber and balsam pear. The FUL-MADS-box transcription factor CsFUL1A inhibits the expression of auxin transporters *PIN7* and *PIN1* by interacting with CsARF12, resulting in reduced auxin accumulation in fruit to regulate cucumber fruit length [81].

Fruit ripening is usually characterized by changes in fruit color, texture, hardness, and an increase in sugar content, one of the most critical physiological processes during fruit development [79]. Members of the ARF family play a role in fruit ripening. For example, it has been confirmed that auxin induces ethylene biosynthesis in apple fruit by activating the expression of *MdARF5* [82]. The interaction between the CpARF2 and the ethylene signal transcription factor CpEIL1 (ETHYLENE-INSENSITIVE3-LIKE1) mediates the interaction between auxin and ethylene signaling to regulate fruit ripening in papaya (*Carica papaya* L.) [83]. In transgenic tomato plants overexpressing *SlARF2*, fruits matured faster than wild-type ones, and enhanced ethylene biosynthesis gene expression and increased ethylene production were detected. [84]. These findings suggest that ethylene and auxin signaling interplay orchestrates fruit ripening.

Recent studies have found that members of the ARF family not only play a role in fruit ripening but also play an important role in fruit coloration. For instance, two ARF2 paralogs (ARF2A and ARF2B) are critical components of the ripening regulatory network in tomatoes. Down-regulation of *ARF2A* and *ARF2B* significantly negatively affected the expression of key ripening genes, thereby inhibiting the ripening of tomato fruit, resulting in the fruit exhibiting reduced pigment accumulation and enhanced firmness [85]. SlARF4 plays a role in fruit ripening mainly by controlling sugar metabolism in tomatoes. Downregulation of *SlARF4* results in a range of ripening-related phenotypes in fruit, such as enhanced fruit firmness and chlorophyll content [86]. Increased chloroplast abundance improves plant nutritional quality and fruit color. Therefore, chloroplast development and photosynthetic activity of green fruit affect the composition and quality of mature tomatoes [87]. Research has shown that SlARF10 plays an essential role in sugar and chlorophyll accumulation during tomato fruit development [88]. Likewise, SlARF6A was found to be involved in fruit development, sugar accumulation, and photosynthesis in tomatoes [89] (Figure 3).

The above data show that auxin mainly regulates the growth process of plant roots or single organs, such as leaves or shoots, through auxin synthesis and homeostasis-related factors, auxin transporter protein family, auxin response factors ARFs and transcriptional repressors Aux/IAAs. There is a recent claim that tomato *Aux⁄IAA* genes play a similar role in maintaining certain vegetative growth processes. Individual members of the Aux/IAA family can participate in different developmental processes. For example, transgenic lines with downregulated expression of *SlIAA15* display pleiotropic phenotypes, including decreased apical dominance, decreased trichomes, altered axillary bud, and xylem development patterns, and increased LR formation [90]. Downregulation of *SlIAA9* in tomatoes has been reported to affect many aspects of vegetative and reproductive growth, including leaf morphology, fruit set and development, and apical dominance [91,92]. Similarly, downregulation of *SlIAA3* resulted in a series of auxin-related and ethylene-related developmental defects, including reduced auxin response, apical dominance, and exaggerated apical hook in etiolated seedlings [93]. These phenotypes provide compelling evidence that *Aux⁄IAA* genes in tomatoes can regulate multiple vegetative growth processes and that different *Aux/IAA* genes can play specific and distinct roles (Figure 2).

On this basis, some factors related to auxin response can play multiple roles not only in the vegetative growth of plants but also in reproductive growth. Down-expression of strawberry (*Fragaria* vesca L.) *FvYUC6* gene and development of roots, leaves, flowers, and fruits of the plant were highly affected. In transgenic strawberries, changes in transcript levels and free IAA levels of the *FvYUC6* gene were closely related to the expression of a subset of auxin-responsive genes. This observation supports the critical role of the gene product in the vegetative and reproductive development of woodland strawberries [94]. Similar results were observed in strawberries (*Fragaria*×ananassa Duch.), and FaYUC1-2 may participate in many developmental processes in strawberries, including flower and fruit development. These observations correspond with the vital role of the *YUC* gene in auxin synthesis, and illustrate the critical importance of auxin for normal plant growth [95] (Figure 3).

## 5. The Role of Auxin in Horticultural Plants under Stress

### 5.1. The Roles of Auxin in Abiotic Stress Tolerance in Horticulture Plants

Recently, many studies have reported that auxins are involved in plant resistance to abiotic stresses. A previous investigation of sunflowers (*Helianthus annuus* L.) showed that the application of exogenous IAA can reduce the toxic effects of lead and zinc on plants and improve the heavy metal resistance of plants, thereby promoting plant growth [96]. Overexpression of *MdIAA24* enhances apple resistance to cadmium (Cd), possibly due to its ability to improve antioxidant capacity and reduce Cd absorption in plants [97]. Previous studies have shown that the regulation of auxin homeostasis is critical for adapting citrus (*Citrus sinensis* Osbeck) to alkaline stress [98]. On the other hand, the overexpression of *MdIAA24* in apples can positively regulate the establishment of the symbiotic relationship between apple roots and arbuscular mycorrhizal fungi (AMF) to cope with drought stress [99]. Furthermore, the expression of *MdGH3-2* and *MdGH3-12* was upregulated during mycorrhizalization, and silencing *MdGH3-2* and *MdGH3-12* negatively affected AM colonization. The root dry weight of the *MdGH3-2/12*-RNAi line was lower under AM inoculation conditions. Compared with wild-type, mycorrhizal transgenic plants were more sensitive to drought stress, and these data suggest that MdGH3-2/12 plays an essential role in apple arbuscular mycorrhizal symbiosis and drought tolerance [100]. Interestingly, MsGH3.6 is a negative regulator of water deficit stress tolerance in apples [101].

Plant adaptation to temperature changes is also modulated by phytohormone signals. In apples, the basic helix–loop–helix transcription factor PHYTOCHROME-INTERACTING FACTOR 4 (MdPIF4) could transactivate the *MdYUCCA8a* promoter, which promoted IAA accumulation, thereby affecting apical dominance and silique malformation under high-temperature conditions [102]. In addition, high temperature also promoted the expression of *GmYUCCA3*, *GmYUCCA5*, and *GmYUCCA7* to increase the accumulation of IAA and promote the elongation of soybean hypocotyl [103]. MYB transcription factors McMYB4 can improve the resistance of apples to temperature changes by upregulating the expression of *AUX/ARF* and *BRI/IN* to promote Auxin and Brassinosteroids signaling pathways [104].

Finally, it has been shown that environmental stress alters auxin distribution and homeostasis mediated by auxin transporters [105]. The expression profiles of *ClLAX, ClPIN,* and *ClABCB* in watermelon (*Citrullus* lanatus) were changed under abiotic stress, which may accelerate or slow down endogenous auxin transport in watermelon seedlings. It suggests auxin redistribution and transportation may be required when watermelons respond to abiotic stress [106] (Figure 4).

### 5.2. The Roles of Auxin in Biotic Stress Tolerance in Horticulture Plants

Auxin is an integral part of the hormone signaling network and is also involved in the response of plants to various pests and diseases. Trichomes can act as a physical or chemical barrier to help plants defend against arthropod pests [107]. A recent study reported that *SlARF4* is highly expressed in type II, V, and VI trichomes and positively regulates auxin-induced formation of type II, V, and VI trichomes in tomato leaves. *SlARF4*-overexpressing plants have a high density of trichomes on the leaf surface, which enhances the plant’s tolerance to red spider mites [61]. Plants may exhibit a susceptible phenotype when pathogens infect plants and disrupt the auxin signaling pathway. Tomato spotted wilt virus (TSWV) causes severe losses of tomato crops worldwide. DNA methylation changes induced by TSWV downregulate *SlARF8* gene expression and disrupt auxin signaling, resulting in tomatoes showing susceptibility to TSWV [108]. ToBRFV (tomato brown rugose fruit virus) is a novel tobacco virus that inhibits tomato root development. In ToBRFV-infected tomato roots, miR160 activity was affected, resulting in abnormal accumulation of *SlARF10a* transcripts. An unusually high accumulation of *SlARF10a* ultimately suppressed root branching in ToBRFV-infected plants [109]. In citrus (*Citrus sinensis* Osbeck), GH3.1 and GH3.1L play essential roles in response to Xanthomonas citri subsp. citri. Overexpression of *CsGH3.1* and *CsGH3.1L* reduces plant susceptibility to citrus canker by enhancing defense responses and inhibiting auxin signaling [110] (Figure 4).

## 6. Conclusions and Perspectives

Recent advances have expanded our understanding of the role of auxins in horticultural plant growth, development, and stress responses. From the above studies, it was further verified that the activity of auxin in various physiological processes is regulated by the following three main regulatory steps: auxin biosynthesis and inactivation, auxin transport, and signal transduction. Among them, various Aux/IAA-ARF signaling factors play their role mainly by activating or inhibiting the expression of downstream auxin-responsive genes from realizing the precise regulation of a particular physiological process. The realization of these three main regulatory steps is inseparable from the role of various transcription factors. There are currently 58 transcription factor families in plants, many of which have numerous members and diverse functions, regulating multiple plant physiological processes; these include the MADS-box, MYB, NAC, and HD-Zip TF families [111]. Different transcription factors are involved in the above three main auxin regulatory steps, regulating various physiological processes of plant growth and development. This shows the complexity and sophistication of auxin-regulated physiological processes.

Generally speaking, phytohormones do not act individually but act synergistically, antagonistically, or additively with other phytohormones or other signaling factors to participate in the regulation of plant physiological processes. In plants, Ca^2+^ is a significant second messenger, and Ca^2+^ signal transduction can regulate various physiological responses, including plant responses to biotic and abiotic stresses, responses to light, and responses to various plant hormones, such as the response to auxin [112,113]. However, the source of Ca^2+^ signaling involves dynamic Ca^2+^ storage by cell surface AGPs [114,115]. AGPs (arabinogalactan proteins) are a class of hydroxyproline-rich glycoproteins that widely participate in plant growth and development. It is universal for plant growth and development, pollen tube formation, and tolerance to abiotic stresses [116]. Lamport and colleagues analyzed AGP as a periplasmic Ca^2+^ capacitor involved in intramolecular Ca^2+^ binding, assuming that dynamic Ca^2+^ is cycled by AGP-Ca^2+^ in many biological processes. They also propose an AGP-Ca^2+^ (secondary messenger)-auxin signaling (primary messenger) cascade model that has the potential to explain the molecular mechanisms by which auxin regulates various physiological processes. In Lampert’s paper, examples of this model participating in essential functions, such as plant cell extension, tropism, mechanical conduction, morphogenesis, and stress response, are listed to explain the rationality of the above hypothesis [117]. It has been reported that the structural and morphological characteristics of AGPs in apples and the calcium concentration in AGPs are related to the fruit ripening process [118]. It seems to be a good start for horticultural researchers to encourage people to explore further whether this model exists in the various physiologies of horticultural plants and lay a foundation for depth analysis of the molecular mechanism of the auxin effect.

This paper lists a large number of studies on the role of auxin in the growth and development and stress response of various horticultural plants to provide ideas for horticultural researchers or horticultural plant cultivators to obtain normal growth, high yield, and high-quality horticultural plants. For example, a large number of studies on horticultural plant fruits can provide a theoretical basis for obtaining high-yield, high-quality, and long-shelf-life fruits and fruit vegetables in the future. The research can apply to the flowering process of horticultural plants to prolong the flowering period of various ornamental flower plants and improve their ornamental value. However, horticultural plant resources are abundant. Many plants have unique physiological structures, such as the leaf heading of Chinese cabbage, the card-shaped inflorescences of cauliflowers, the numerous spines on the surface of cucumber fruit, and the irregular bulges on the surface of the bitter gourd. Only a few researchers have paid attention to these unique horticultural traits, and their formation mechanism has not been analyzed. Whether auxin is involved in the formation mechanisms of these unique horticultural traits is also important content to further promote the functional analysis of auxin. It offers a new perspective for auxin-related research in the future.

## Figures and Tables

**Figure 1 cells-11-02761-f001:**
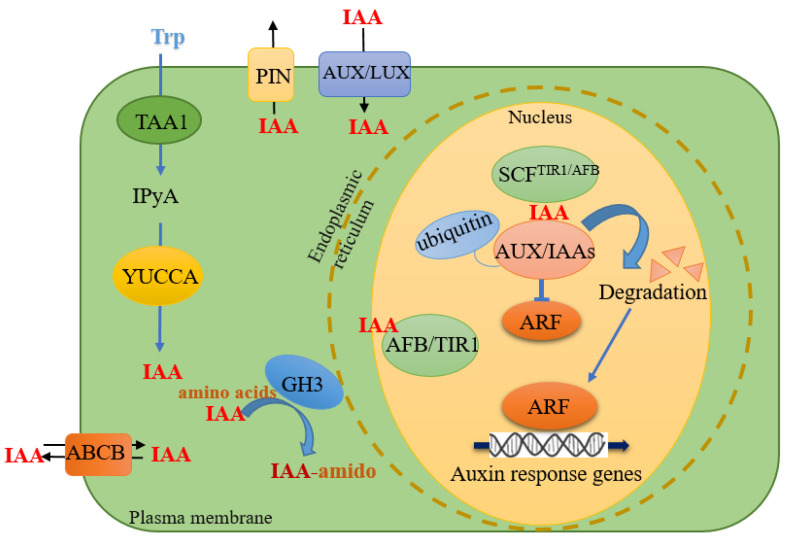
An abbreviated cellular model of auxin biosynthesis, transport, and signaling. IAA is biosynthesized from Trp via IPyA by TAA1 and YUCCA in the TAA/YUC pathway. IAA is transported by AUX1/LAX1, PIN, and ABCB proteins. The auxin-induced GH3 enzyme converts active IAA to inactive IAA-amido. Auxin induces ubiquitination and degradation of Aux/IAA proteins via SCF^TIR1/AFB^. Degradation of the Aux/IAA repressor recovers ARF activity to activate transcription of auxin-responsive genes.

**Figure 2 cells-11-02761-f002:**
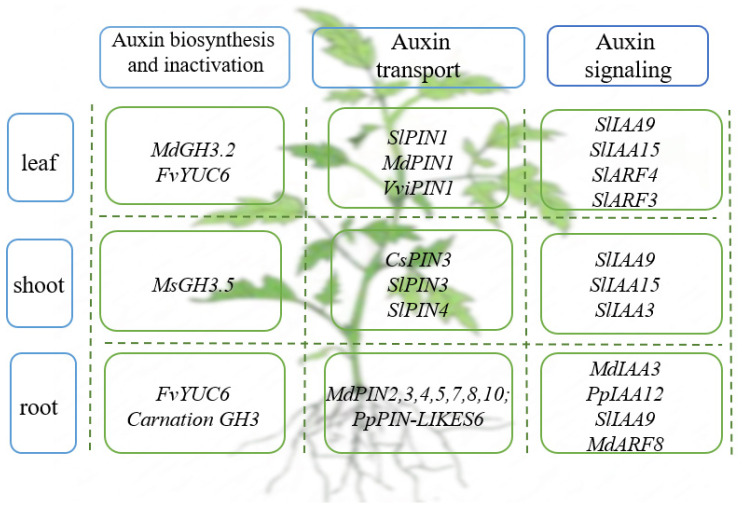
Auxin-related genes that play a role in the vegetative growth of horticultural plants.

**Figure 3 cells-11-02761-f003:**
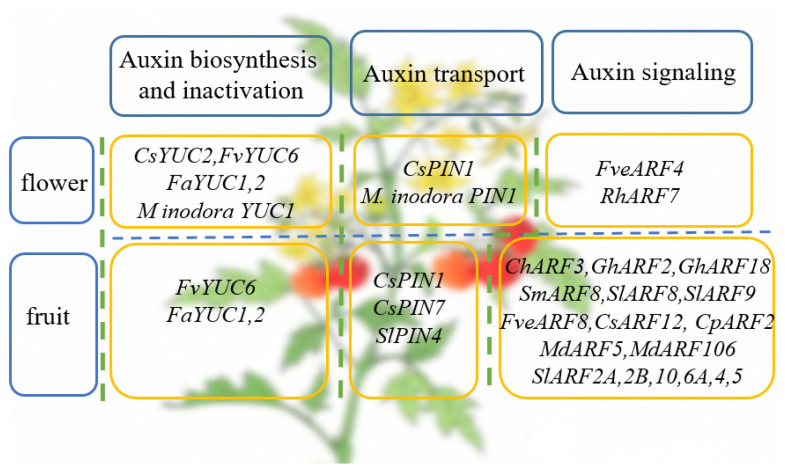
Auxin-related genes that play a role in the reproductive growth of horticultural plants.

**Figure 4 cells-11-02761-f004:**
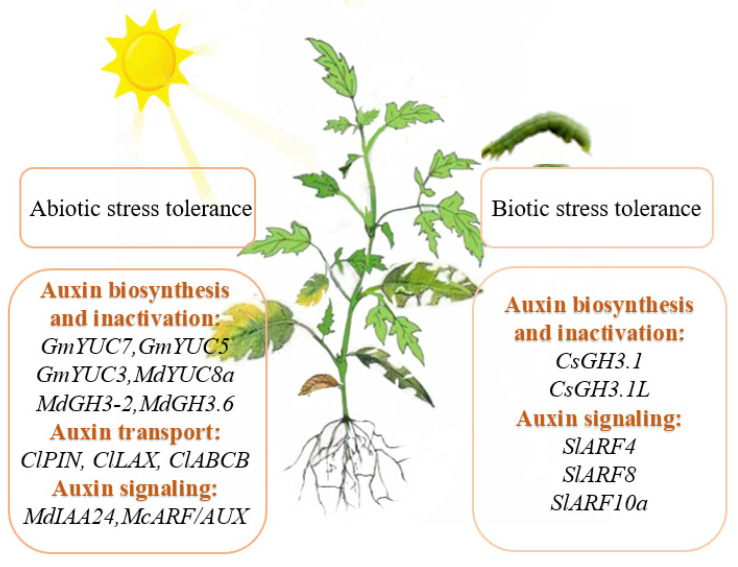
Auxin-related genes that play a role in stress tolerance of horticultural plants.

## Data Availability

Not applicable.

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
