# Peer review of "Roles of Auxin in the Growth, Development, and Stress Tolerance of Horticultural Plants"

_cells, 2022, doi:10.3390/cells11172761_

Round 1

Reviewer 1 Report

The reviewed article summarizes reports on involvement of auxins in the control of growth, development and resistance to biotic and abiotic stresses of horticulture plants. It may be interesting to readers, since horticulture plants did not receive enough attention in this regard. Still I recommend revision of the article.

1.       I think that some important information is missing in the article. It addresses mechanisms of auxin metabolism and signaling and their relation to the growth of plant organs, while I failed to find information on target auxin genes and processes on cellular level. Information on the effects of these hormones on cell division, extension and differentiation should be added. Auxins effects on vasculature development are only mentioned, while they are most important for the control of, e.g., leaf shape, which is one of the main themes of the present review.

2.       I advise to add a table summarizing mechanisms involved in auxin synthesis, conjugation, transport and signaling. This would help authors not to repeat the same description of factors involved in them and readers will be able to find what is meant when those factors are mentioned numerous times in the text.

3.       English of the text should be improved. I am going to attach the file of the article where I shall mark those sentences that need to be revised. I shall also explain what is likely to be wrong with them. Since English is not my first language I recommend that the authors edit it with the help of professional translators.

4.      Authors describe importance of adventitious roots for adaptation to waterlodging. If they want to address involvement of roots in adaptation to this stress factor, they should at least also add importance of root architecture for adaption to water deficit.

5.       Some sections start with the same description of auxin functions that were mentioned in the introduction. I got the impression that the article was written by several authors who did not read other parts of the text except their own.

My other comments are arranged according to the position in the text of the sentences they are related to.

6.       Line 118. “up-regulated during AR prolongation” – I think it should be elongation. Prolongation is mentioned when it is meant that AR are prongation of other organs.

7.       “Post-translational modification of ARFs and IAAs” – IAA means auxin itself (indole acetic acid), which cannot be post-translationally modified. I think that authors meant Aux/IAA.

8.       PpSnRK1 is improperly deciphered as “sucrose non-INDUCTION-1-protein kinase-1.” It should be NON-FERMENTING-1-RELATED protein kinase.

9.       When mentioning SUMO and and sumoliation it should be explained that sumoylation involves ligases similar to ubiquitinylation and prepares proteins for processes such as subcellular localization

10.   Line 147. “Auxin inhibits CK biosynthesis during this process” – I failed to find this information in the cited reference [39]. Authors should check this and if they think this information important, they should find another reference.

11.     Lines 152-153. “In tomato, inhibition of tomato SlPIN4 and SlPIN3 expression altered shoot structure”- I think it relevant to mention here that preventing auxin export from dormant buds by inhibition of PINs, would block their outgrowth (with reference to [38])

12.   I think it should be mentioned that MdbHLH3 means basic helix-loop-helix transcription factor

13.   Lines 197-198. “cystatin-like protein DEFORMED FLORAL BUD1 (CsDFB1) was found to function as a transcriptional regulator, inhibiting the expression of CsYUC2 and CsPIN1” – this citation is incorrect. It is said in cited article [61] that it acts in an opposite way on CsYUC2 and CsPIN1:“CsDFB1 impaired the CsPHB-dependent ACTIVATION of CsYUC2 and the REPRESSION of CsPIN1.

14.   When mentioning amiRNA I advise to decipher that it means artifical microRNA. The meaning of the term microRNA silencing should be also explained to make it clearer what miRNA technique results in.

15.    When mentioning CpEIL1, I advise to explain that it means ethylene signal transcription factor

16.    MdPIF4 I advise to explain that it means PHYTOCHROME-INTERACTING FACTOR

17.     When mentioning McMYB4, I advise to add that this is a transcription factor.

18.    Line 335. Relation of SlMYB75 to auxins seems unclear to me.

Reviewer 2 Report

The abstract states: Auxin regulates virtually every aspect of plant growth and development with the aim summarised at line 394: to further promote the functional analysis of auxin to carry out auxin-related research in the future.

While this is a comprehensive review of plant biology the final sentence of the review shows that functional analysis remains as a huge gap, with no explanation of auxin mechanism at the molecular level to explain how auxin can be involved in virtually every aspect of plant biology!

However, in 2013 Two papers appeared that shed light on the problem; Firstly, Vanneste & Friml suggested that calcium is the missing link in auxin action; secondly, discovery of specific Ca2+ binding by cell surface arabinogalactan proteins (AGPs) Lamport & Varnai, identified AGPs as the missing link between the auxin-activated proton pump and the source of  dynamic cytosolic Ca2+.

Those papers and even more recent publications go some way to account for the ubiquitous involvement of auxin in plant biology. Perhaps a mention of these molecular aspects would be a suitable conclusion to the review. It also raises the question of so-called “acid growth” not mentioned in the review...probably because a direct effect of low pH on the cell wall remains unresolved! However, an auxin-activated ATPase proton pump of the plasma membrane plays three crucial roles which account for the ubiquity of auxin effects:

1.      It generates a membrane potential of ca. -160mV that enhance cation influx and anion efflux.

2.      It enables release of Ca2+ from AGP-Ca2+ the source of cytosolic Ca2+.

3.      It promotes auxin transport by protonating anionic auxin thus allowing neutral auxin to traverse the lipid plasma membrane for influx into adjacent cells.

This interesting paper needs a little minor editing of a redundant phrase at at lines 230 and 231.

Round 2

Reviewer 1 Report

Authors addressed all my comments and changed the text in accordance. I am satified and think that it may be published in its present state

Author Response

Response to the Reviewer’s comments

Reviewer #1 (Comments for the Author):

Authors addressed all my comments and changed the text in accordance. I am satified and think that it may be published in its present state.

Reply: Thank you very much for your affirmation of this manuscript, and thank you again for your valuable comments which are of great significance to improving this manuscript.